# Long-Term Pulmonary Sequelae and Immunological Markers in Patients Recovering from Severe and Critical COVID-19 Pneumonia: A Comprehensive Follow-Up Study

**DOI:** 10.3390/medicina60121954

**Published:** 2024-11-27

**Authors:** Edita Strumiliene, Jurgita Urbonienė, Laimute Jurgauskiene, Ingrida Zeleckiene, Rytis Bliudzius, Laura Malinauskiene, Birutė Zablockiene, Arturas Samuilis, Ligita Jancoriene

**Affiliations:** 1Clinic of Infectious Diseases and Dermatovenerology, Faculty of Medicine, Institute of Clinical Medicine, Vilnius University, 01513 Vilnius, Lithuania; birute.zablockiene@santa.lt; 2Centre of Infectious Diseases, Vilnius University Hospital Santaros klinikos, 08661 Vilnius, Lithuania; jurgita.urboniene@santa.lt; 3Clinic of Cardiac and Vascular Diseases, Faculty of Medicine, Institute of Clinical Medicine, Vilnius University, 01513 Vilnius, Lithuania; laimute.jurgauskiene@santa.lt; 4Department of Radiology, Nuclear Medicine and Medical Physics, Faculty of Medicine, Institute of Biomedical Sciences, Vilnius University, 01513 Vilnius, Lithuania; ingrida.zeleckiene@santa.lt (I.Z.); rytis.bliudzius@santa.lt (R.B.); arturas.samuilis@santa.lt (A.S.); 5Clinic of Chest Diseases, Immunology and Allergology, Faculty of Medicine, Institute of Clinical Medicine, Vilnius University, 01513 Vilnius, Lithuania; laura.malinauskiene@santa.lt

**Keywords:** COVID-19, long COVID, pulmonary function, lymphocytes, CCR2, chest CT

## Abstract

*Background and Objectives*: Severe and critical COVID-19 pneumonia can lead to long-term complications, especially affecting pulmonary function and immune health. However, the extent and progression of these complications over time are not well understood. This study aimed to assess lung function, radiological changes, and some immune parameters in survivors of severe and critical COVID-19 up to 12 months after hospital discharge. *Materials and Methods*: This prospective observational cohort study followed 85 adult patients who were hospitalized with severe or critical COVID-19 pneumonia at a tertiary care hospital in Vilnius, Lithuania, for 12 months post-discharge. Pulmonary function tests (PFTs), including forced vital capacity (FVC), forced expiratory volume in 1 s (FEV1), and diffusion capacity for carbon monoxide (DLCO), were conducted at 3, 6, and 12 months. High-resolution chest computed tomography (CT) scans assessed residual inflammatory and profibrotic/fibrotic abnormalities. Lymphocyte subpopulations were evaluated via flow cytometry during follow-up visits to monitor immune status. *Results*: The median age of the cohort was 59 years (IQR: 51–64). Fifty-three (62.4%) patients had critical COVID-19 disease. Pulmonary function improved significantly over time, with increases in FVC, FEV1, VC, TLC, and DLCO. Residual volume (RV) did not change significantly over time, suggesting that some aspects of lung function, such as air trapping, remained stable and may require attention in follow-up care. The percentage of patients with restrictive spirometry patterns decreased from 24.71% at 3 months to 14.8% at 12 months (*p* < 0.05). Residual inflammatory changes on CT were present in 77.63% at 6 months, decreasing to 69.62% at 12 months (*p* < 0.001). Profibrotic changes remained prevalent, affecting 82.89% of patients at 6 months and 73.08% at 12 months. Lymphocyte counts declined significantly from 3 to 12 months (2077 cells/µL vs. 1845 cells/µL, *p* = 0.034), with notable reductions in CD3+ (*p* = 0.040), CD8+ (*p* = 0.007), and activated CD3HLA-DR+ cells (*p* < 0.001). This study found that higher CD4+ T cell counts were associated with worse lung function, particularly reduced total lung capacity (TLC), while higher CD8+ T cell levels were linked to improved pulmonary outcomes, such as increased forced vital capacity (FVC) and vital capacity (VC). Multivariable regression analyses revealed that increased levels of CD4+/CD28+/CD192+ T cells were associated with worsening lung function, while higher CD8+/CD28+/CD192+ T cell counts were linked to better pulmonary outcomes, indicating that immune dysregulation plays a critical role in long-term respiratory recovery. *Conclusions*: Survivors of severe and critical COVID-19 pneumonia continue to experience significant long-term impairments in lung function and immune system health. Regular monitoring of pulmonary function, radiological changes, and immune parameters is essential for guiding personalized post-COVID-19 care and improving long-term outcomes. Further research is needed to explore the mechanisms behind these complications and to develop targeted interventions for long COVID-19.

## 1. Background

The global COVID-19 pandemic, caused by the SARS-CoV-2 virus, has led to significant morbidity and mortality since its emergence in late 2019. While most individuals experience mild to moderate symptoms, a subset of patients develops severe or critical illness, often resulting in acute respiratory distress syndrome (ARDS). This severe form of COVID-19 frequently necessitates high-flow oxygenation or mechanical ventilation in intensive care settings [1,2]. The underlying pathophysiology of severe COVID-19 involves extensive pulmonary inflammation and damage, which may lead to enduring respiratory complications even after the resolution of the acute infection [1,2,3].

Survivors of severe and critical COVID-19 pneumonia are at heightened risk for chronic pulmonary sequelae, including interstitial lung damage with pulmonary fibrosis and persistent pulmonary function abnormalities. Studies have documented that post-acute sequela of SARS-CoV-2 infection (PASC), commonly referred to as long COVID or post-COVID, often include sustained respiratory symptoms such as dyspnoea, cough, decreased exercise capacity, and changes in chest CT [1,4]. Long-term impacts on lung function and structure, measured by pulmonary function tests (PFTs) and radiological imaging and ongoing pathophysiological mechanisms degerming long COVID symptoms, remain an essential area of ongoing research.

Beyond pulmonary complications, severe COVID-19 also exerts profound effects on the immune system. Dysregulated immune responses, including lymphopenia and altered T-cell function, have been commonly observed in critically ill patients [5]. Lymphocytes, especially T cells, play a vital role in viral clearance and the regulation of the immune response. However, in severe COVID-19, these cells often become dysfunctional, contributing to the prolonged inflammation and immune dysregulation seen in long COVID-19 [5].

One immune marker of particular interest is the C-C chemokine receptor 2 (CCR2 or CD192, cluster of differentiation 192), a receptor expressed on various immune cells, including T cells. CCR2 is crucial for regulating cell activation and migration during infection. Studies have suggested that CCR2-dependent signaling may play a role in the development of lung fibrosis, with inhibition of this pathway shown to reduce fibrotic changes in experimental models [6,7]. Understanding the role of CCR2 in the context of post-COVID-19 immune dysregulation may provide new insights into potential therapeutic targets.

Long-term studies have highlighted that patients with severe initial COVID-19 are at higher risk for persistent abnormalities, including pulmonary fibrosis and immune dysfunction. However, independent biomarkers that could predict these outcomes are still not fully understood [3]. This study seeks to explore the longitudinal changes in lung function, radiological findings, and immunological markers over a 12-month period in patients recovering from severe or critical COVID-19 pneumonia. By identifying key predictors of poor long-term outcomes, we aim to contribute to the development of personalized post-COVID management strategies.

## 2. Study Rationale and Objectives

This study aims to provide a detailed evaluation of pulmonary and immunological outcomes in patients who survived severe or critical COVID-19 pneumonia. By conducting follow-up assessments at 3, 6, and 12 months post-discharge, we seek to

Characterise the progression of lung function recovery or deterioration using pulmonary function tests (PFTs);Analyse the persistence and resolution of radiological abnormalities through chest CT scans;Investigate the longitudinal changes in immunological markers, including lymphocyte subpopulations, and their association with lung function and radiological findings;Identify potential predictors of long-term pulmonary impairment and immune dysregulation to inform clinical management and rehabilitation strategies for COVID-19 survivors.

## 3. Materials and Methods

### 3.1. Patients

All adult patients with a real-time reverse transcriptase polymerase chain reaction (RT-PCR) test–confirmed SARS-CoV-2 infection and severe or critical COVID-19 pneumonia, treated at Vilnius University Hospital “Santaros klinikos”, a tertiary hospital, from October 2021 to October 2022, were offered to participate in this prospective follow-up study at the time of discharge if they met the following inclusion criteria:Radiologically confirmed lung injury via chest X-ray and/or CT scan at the onset of the disease;Willingness and ability to complete pulmonary function tests and to undergo chest CT and blood testing at the follow-up visits;No chronic lung disease prior to the infection (to avoid radiological and functional overdiagnosis);Ability to understand and sign the informed consent to participate in this study.

Patients with mild or moderate COVID-19, according to the WHO classification, were not included in this study, as they did not have severe lung injury and were at a lower risk for residual lung damage and long COVID-19 respiratory symptoms. Known immunosuppressive conditions were also considered as an exclusion criterion.

The follow-up visits were organized at the Respiratory Outpatient Department at 3, 6, and 12 months after discharge. Eighty-five consecutive eligible patients were included.

The patients included in this study were categorised into the following groups according to WHO criteria [6]:Severe disease: characterised by radiological evidence of bilateral pneumonia with lung injury > 50% and any of the following: respiratory rate ≥ 30 breaths/min; oxygen saturation ≤ 93% at rest; oxygen therapy ≤ 10 L/min; no need for treatment in the Intensive Care Unit (ICU);Critical disease: characterised by respiratory failure requiring high-flow oxygen therapy or intubation, shock, or other organ failure necessitating ICU care.

### 3.2. Data Collection

Medical records of the participants were reviewed from the hospital database, and demographic data and chest radiological data at the time of hospitalization (acute disease) were analysed.

Patients were evaluated at three visits: 3, 6, and 12 months after discharge from the hospital. During these visits, lung function tests and blood samples were collected and analysed. Chest CT scans were performed at 6 and 12-month visits. If pathological findings were not observed during a test, that test was not repeated in subsequent visits.

This study was approved by the Vilnius Regional Biomedical Ethics Committee (Protocol number: 2020/6-1233-718; 2020-06-22) and conducted in accordance with the ethical standards specified in the Declaration of Helsinki. All patients provided informed written consent to participate in this study.

#### 3.2.1. Pulmonary Function Testing

The pulmonary function tests were performed using Vmax Encore (Viasys^®^ Healthcare, Conshohocken, PA, USA). The following parameters were measured: forced vital capacity (FVC), forced expiratory volume in the first second of exhalation (FEV1), FVC/FEV1 ratio, total lung capacity (TLC), vital capacity (VC), residual volume (RV), and diffusion capacity of the lung for carbon monoxide (DLCO), measured using the single-breath test. The DLCO was corrected for haemoglobin levels. All parameters were expressed as percentages of the predicted normal value; the lower limits of normal (LLN) were taken into account.

#### 3.2.2. Chest CT Protocols and Image Analysis

At the time of admission, all patients with suspected COVID-19 pneumonia underwent posterior–anterior and lateral chest X-rays or chest CT examinations according to the decision of the attending physician. The CT scans were performed during follow-up visits. For the portion of our cohort who did not undergo a CT scan at the initial stage of the disease, posterior–anterior and lateral chest X-rays were used to assess the extent of the disease.

All chest CT scans were performed using two helical CT scanners: (1) a 64-row GE Discovery CT750 HD (General Electric Healthcare, Waukesha, WI, USA). A low-dose lung CT protocol was used. All CT images were acquired at the end of inhalation. Acquisition parameters were set as follows: tube voltage 100 kV, automatic tube current modulation, SmartmA/AutomA, Large Body FOV, detector coverage 40 mm, slice thickness 3.75 mm, pitch 1.375, and gantry rotation time 0.4 s. All images were then reconstructed with a slice thickness of 1.25 mm using lung and standard algorithms. Adaptive Statistical Iterative Reconstruction (ASiR) with 30% mA reduction was used to reduce the X-ray dose. (2) BrainLab Airo mobile CT (Manufacturer Mobius Imaging, LLC., Shirley, MA, USA) was also used for chest CT. Acquisition parameters were set as follows: tube voltage 120 kV, tube current 50 mA, rotation time 1.92 s, Large Body FOV, detector coverage 40 mm, slice thickness 1.25 mm, and pitch 1.415. All images were reconstructed using lung and standard algorithms.

Image analysis was performed by two independent radiologists. For studies where a discordance of scoring was found, further review was performed by a third radiologist until a consensus was reached.

CT features such as ground-glass opacities (GGOs), consolidation, parenchymal bands, architectural distortion, air bronchograms, and bronchiectasis, as well as their distribution were noted. The findings were classified into two groups: inflammatory changes (including GGO and consolidation) and fibrotic/reticular changes (including parenchymal bands, architectural distortion, and bronchiectasis). The involvement of each lung lobe was quantified using a method previously employed in other studies evaluating pulmonary fibrosis caused by SARS [7,8]. Each of the five lung lobes was assigned a score of 0–5 points for inflammatory and fibrotic/reticular changes. Points were assigned as follows: 0 for a lobe without perceptible changes, 1 for lesions involving up to 5% of a lobe, 2 for lesions involving 6–25% of a lobe, 3 for lesions involving 26–50% of a lobe, 4 for lesions involving 51–75% of a lobe, and 5 for lesions involving more than 75% of a lobe [2,3]. In the analysis of images taken during the acute COVID-19 period, the scores for each lobe were summed, with a maximum total score of 25, and one score was attributed to the extent of lesions, irrespective of the type of CT features. In contrast, during follow-up, two separate scores were attributed to inflammatory and fibrotic changes, and a total radiological score was determined.

#### 3.2.3. Blood Samples

For lymphocyte population analysis, peripheral blood samples were stained with monoclonal antibodies targeting surface markers CD3, CD4, CD8, CD19, HLA-DR, CD28 (BD Biosciences, Franklin Lakes, NJ, USA), and CD192 (BioLegend, San Diego, CA, USA). After incubating in the dark at room temperature, red blood cells were lysed using FACS lysing solution and washed with phosphate-buffered saline. Following fixation, these labelled cells were acquired on a FACSCalibur flow cytometer (BD Biosciences, Franklin Lakes, NJ, USA) using CellQestPro analysis software (Version 5.2.1) after proper instrument settings, calibration, and compensation. Analysis regions were gated using a CD45/CD14 antibody combination, and an isotype control γ1/γ2a was used for negative marker settings. Absolute numbers of lymphocyte subsets were calculated using the absolute lymphocyte counts obtained from the haematology analyser and adjusted to specific subset percentages.

#### 3.2.4. Statistical Analysis

Statistical analysis was performed using IBM SPSS version 20.0. GraphPad Prism was used to create graphics. The Shapiro–Wilk test was used to test the normality of data; the data did not exhibit a normal distribution. Continuous and categorical variables are presented as median (interquartile range (IQR)) or mean (±standard deviation (SD)) and as numbers (percentages), respectively. The Mann–Whitney U test and the Wilcoxon Signed-Rank test for paired samples were used to compare continuous variables, while the χ^2^ test or Fisher’s exact test was used to compare categorical variables. A multivariable linear regression model was created to determine the association between pulmonary function parameters and immunological markers. The model included pulmonary function parameter or CT change severity score as the dependent variable and age, gender, lymphocyte count, and lymphocyte subpopulation count as predictors. A multivariable binary logistic regression model was applied to determine the association between a restrictive spirometry pattern and immunological markers. The model included the presence of a restrictive spirometry pattern as the dependent variable, and age, gender, COVID-19 severity, lymphocyte count, and lymphocyte subpopulation count as predictors. The backward conditional method was used for multivariable binary logistic regression. A two-sided *p*-value of <0.05 was considered significant.

## 4. Results

A total of 85 patients were included in the analysis, comprising 40 (47.1%) women and 45 (52.9%) men. The median age was 59 years (IQR 51–64). Among these patients, 53 (62.4%) had recovered from critically severe COVID-19, and 35 (37.6%) had recovered from severe COVID-19.

### 4.1. Pulmonary Function Tests

Pulmonary function test results at 3, 6, and 12 months post-discharge are provided in Table 1. The pulmonary function test results showed significant improvement in lung function over time. Forced vital capacity (FVC), forced expiratory volume in one second (FEV1), and vital capacity (VC) all improved between 3, 6, and 12 months post-discharge (*p* < 0.05). The FEV1/FVC ratio remained largely unchanged and was within the range of normality in the majority of the cases.

At 3 months, 24.71% of patients showed restrictive patterns in spirometry, with the rate statistically significantly decreasing to 20.24% at 6 months and 14.8% at 12 months (Figure 1).

### 4.2. Radiological Findings

A total of 76 out of 85 patients returned for the 6-month CT evaluation, and 79 returned for the 12-month radiological evaluation. Fibrotic/reticular changes were assessed in 78 of these patients. A total of 77.63% of patients still had residual inflammatory changes on CT at 6 months after the discharge. The lower lobes of the lungs were predominantly affected: the right lower lobe (RLL) in 73.68% of patients and the left lower lobe (LLL) in 69.74%. Additionally, inflammatory changes were observed in the right upper lobe (RUL) in 64.47% of patients, the left upper lobe (LUL) in 64.47%, and the right middle lobe (RML) in 53.95%. The total CT inflammatory change severity score was 8 (IQR 1–14).

A total of 69.62% of patients had residual inflammatory changes on CT at 12 months post-COVID-19. The RLL was affected in 64.56% of patients, the RML in 48.10%, the RUL in 53.16%, the LLL in 60.76%, and the LUL in 54.43%. The total CT inflammatory change severity score at 12 months was 6 (IQR 0–12), which was statistically significantly lower compared to the 6-month CT inflammatory change severity score (*p* < 0.001).

A total of 82.89% of patients had fibrotic/reticular changes on CT at 6 months post-discharge. The lower lobes of the lungs were mainly affected: RLL in 75% of patients and LLL in 69.74%. The RUL was affected in 64.47% of patients, the LUL in 69.74%, and the RML in 53.95%. The total CT fibrotic/reticular change severity score was 6 (IQR 2–11).

A total of 73.08% of patients had fibrotic/reticular changes on CT at 12 months post-COVID-19. The RLL was affected in 66.67% of patients, the RML in 51.28%, the RUL in 51.28%, the LLL in 61.54%, and the LUL in 56.41%. The total CT fibrotic/reticular change severity score at 12 months was 4.5 (IQR 0–10), which was statistically significantly lower compared to the 6-month CT inflammatory change severity score (*p* < 0.001).

The representative radiological changes over time are shown in Figure 2.

CT inflammatory change severity scores indicated that lobe lesions at 6 and 12 months were mostly mild, reaching 1–2 severity points, except for the RLL, where lesions were moderate, involving more than 25% of the lobe in 46.05% of patients at 6 months and in 21.52% of patients at 12 months. These scores statistically significantly decreased at 12 months compared to 6 months for the RUL, RML, RLL, and LLL. CT fibrotic/reticular change severity scores were mild and decreased at 12 months compared to 6 months for the RUL, RLL, LUL, and LLL (Table 2).

### 4.3. Immunological Markers

White blood cell (WBC), lymphocyte, and lymphocyte subpopulation counts were performed for 84 patients at 3 months, 65 patients at 6 months, and 59 patients at 12 months. CD4+/28+/192+ counts were conducted for 81 patients at 3 months, 59 patients at 6 months, and 55 patients at 12 months, while CD8+/28+/192+ counts were performed for 81 patients at 3 months, 60 patients at 6 months, and 55 patients at 12 months. WBC, lymphocyte, and lymphocyte subpopulation counts at 3, 6, and 12 months are shown in Table 3.

The WBC count did not change significantly between months 3, 6, and 12 post-discharge.

However, a statistically significant decrease in lymphocyte count, CD3+ cells, and CD8+ cells was observed at month 12 compared to month 3 post-COVID-19.

Additionally, CD3HLA-DR+ cells decreased significantly in month 12 compared to both month 3 and month 6. (Table 3, Figure 3).

### 4.4. Association of Pulmonary Function Tests, Radiological Findings, and Immunological Markers

Patients with a restrictive spirometry pattern had higher scores for CT inflammatory and fibrotic/reticular changes at acute COVID-19 and at 6 and 12 months post-discharge compared to those without a restrictive spirometry pattern, although the differences were not statistically significant. However, patients with a restrictive spirometry pattern at 3 and 6 months had higher CT inflammatory and fibrotic/reticular change severity scores at 12 months compared to those without restriction in spirometry (Table 4).

At 6 months, patients with a restrictive spirometry pattern had a higher CD19+ cell count compared to those without restriction (259.90 cells/μL vs. 171.12 cells/μL, *p* = 0.025). Similarly, at 12 months, patients with a restrictive spirometry pattern had a higher CD19+ cell count compared to those without restriction (279.12 cells/μL vs. 173.74 cells/μL, *p* = 0.042).

Multivariable regression was applied to evaluate the association between pulmonary function tests and the levels of immune cell subsets.

At 3 months, higher levels of CD8+/28+/192+ were associated with a slight decline in pulmonary function, as measured via the FEV1/FVC ratio: for each increase of 1 × 10^9^/L in CD8+/28+/192+, the FEV1/FVC ratio decreased by 0.01. Increased levels of CD4+/28+/192+ were associated with lung function worsening, specifically reduced TLC at 6 months and decreased FVC, FEV1, and VC at 12 months, while increased CD8+/28+/192+ levels were associated with improved lung function (increased FVC, FEV1, and VC at 12 months). Additionally, higher levels of CD3HLA-DR+ were negatively associated with FEV1 and VC at 12 months, suggesting a decline in lung capacity (Table 5). Female gender was associated with a lower DLCO at 3, 6, and 12 months. Analysis revealed that the severity score of inflammatory changes and reticular/fibrotic changes on CT were associated with older age at 12 months (Table 5).

A decreased lymphocyte count (0.99 (95% CI 0.98–1.00, *p* = 0.017), increased CD19+ cell count (1.02 (95% CI 1.00–1.03), and increased CD4+ cell count (OR 1.01 (95% CI 1.00–1.02) were associated with a restrictive spirometry pattern at 12 months.

Patients with residual inflammatory changes on CT had a lower count of CD3HLA-DR+ cells compared to those without inflammatory changes at both 6 and 12 months. At 6 months, the count of CD3HLA-DR+ cells was 367.54 vs. 606.06, *p* = 0.048, and the percentage was 17% vs. 24%, *p* = 0.014. At 12 months, CD3HLA-DR+ cell count was 261.14 vs. 539.15, *p* = 0.021, and the percentage was 14% vs. 19.5%, *p* = 0.024.

## 5. Discussion

This study offers a comprehensive evaluation of the long-term pulmonary and immunological outcomes in patients who survived severe or critical COVID-19 pneumonia, with follow-up assessments over a 12-month period. Our findings reveal that survivors often experience persistent pulmonary sequelae, including impairments in pulmonary function, particularly with the diffusion capacity of the lung for carbon monoxide (DLCO). Although there were gradual improvements in forced vital capacity (FVC), forced expiratory volume in one second (FEV1), residual volume (RV), and vital capacity (VC), complete lung function recovery remained elusive, underscoring the need for extended monitoring and rehabilitation.

Our data showed that female gender was associated with lower DLCO at 3, 6, and 12 months. Additionally, the analysis revealed that higher severity scores for inflammatory and reticular/fibrotic changes on CT at 12 months were linked to older age. These findings are consistent with existing literature, which identifies female gender and older age as risk factors for the development of long COVID [9,10].

The persistence of pulmonary abnormalities observed in our cohort aligns with previous studies on the post-COVID-19 respiratory sequelae [11,12,13]. For instance, Sonnweber et al. (2023) reported long-lasting pulmonary impairments, particularly in diffusion capacity, which persisted beyond 6 months after the discharge [11]. Similarly, Bocchino et al. (2022) demonstrated that 12 months post-COVID-19, many patients continued to exhibit fibrotic changes in their lung tissue, although without significant progression [12]. In contrast, Carvalho et al. (2024) [14] noted a decline in the prevalence of severe pulmonary impairments over a two-year period.

However, our data indicate a higher prevalence of residual changes at 12 months compared to some reports, although without evidence of progressive deterioration during this period [14]. The stable yet persistent fibrotic changes observed in the lower lung lobes suggest long-term structural damage, which is consistent with other findings in the literature [11,12,13,14].

Our study also demonstrated that the severity of the disease is associated with the intensity of radiological abnormalities in the lungs. Critical COVID-19 survivors showed more persistent radiological abnormalities during follow-up with higher scores of any abnormalities, including profibrotic/fibrotic changes, over time compared to those with non-severe disease. These findings are consistent with data published in a systematic review and meta-analysis by M. Babar et al. [15], showing that radiological lung changes like bronchiectasis, reticulation, and other fibrotic-like changes are highly associated with the severity of COVID-19.

Our study highlights significant long-term immunological impacts of severe COVID-19. There were significant decreases in total lymphocyte counts, including CD3+, CD8+, and CD3HLA-DR+ cells, over the 12-month follow-up period. CD4+ T cells remained relatively stable, whereas CD8+ T cells and CD3HLA-DR+ cells showed notable reductions.

This study indicates that elevated levels of CD4+ T cells, particularly the CD4+/28+/192+ subset, correlate with poorer lung function outcomes post-COVID-19. These cells are markers of immune activation and exhaustion. Increased CD4+ T cell counts are associated with reduced TLC and decreased FVC and FEV1. The persistence of high CD4+ T cell counts suggests ongoing inflammation or delayed lung tissue repair, which could contribute to structural lung changes and the exacerbation of respiratory impairment.

Conversely, CD8+ T cells appear beneficial for lung function recovery. Elevated CD8+/28+/192+ counts were positively associated with improved pulmonary parameters, such as FVC, FEV1, and VC. This suggests a potentially protective role of CD8+ T cells in mitigating fibrosis or inflammatory damage. CD8+ T cells likely contribute to a balanced immune response, aiding in the resolution of inflammation without inducing excessive fibrosis or structural changes in lung tissues.

These changes reflect ongoing immune dysregulation over 12 months and are consistent with other studies reporting prolonged alterations in lymphocyte subsets and immune activation post-COVID-19 [9].

Persistent symptoms and immune dysregulation after COVID-19 have been investigated by other authors. Carfì et al. (2020) documented persistent symptoms in patients after acute COVID-19, noting long-term immune alterations similar to our findings. Their study highlighted that lymphopenia and altered T-cell function were common among survivors [1]. Huang et al. (2021) reported that 6-month follow-ups showed sustained immune dysregulation with reductions in lymphocyte counts, similar to those observed in our cohort [2]. Chen et al. (2020) also found that severe COVID-19 is associated with significant immune dysregulation, with decreased CD3+ and CD8+ T cells, which aligns with our results of ongoing lymphocyte reduction [16].

Long-term immune alterations were investigated by Mathew et al. (2020) [17]. They identified distinct immunotypes in COVID-19 patients, with severe cases showing prolonged immune activation and altered T-cell profiles. This is consistent with our findings of reduced CD3+, CD8+, and CD3HLA-DR+ cells over the follow-up period. Davitt et al. (2022) highlighted prolonged immune dysregulation in survivors, noting associations between lymphopenia, altered T-cell function, and ongoing inflammatory responses [5].

Sharif-Zak et al. (2022) and Ranjbar et al. (2022) explored the role of immune markers like CCR2 in COVID-19 severity and recovery, showing that dysregulated immune responses can contribute to prolonged pulmonary issues [18,19]. Our findings support this by demonstrating that specific immunological markers (e.g., CD4+/28+/192+) are associated with long-term pulmonary impairment. Milger et al. (2017) and Ranjbar et al. (2022) discussed the role of regulatory T cells in lung fibrosis, which corresponds to our observation of stable fibrotic changes and their correlation with immune dysregulation over time [19,20].

Our study’s findings align closely with existing literature, reinforcing the understanding that severe COVID-19 leads to significant and persistent immune dysregulation. The observed reductions in lymphocyte counts, particularly CD3+, CD8+, and CD3HLA-DR+ cells, are consistent with other studies that highlight prolonged immune activation and altered T-cell profiles in COVID-19 survivors. The association between elevated CD4+/28+/192+ cell counts and impaired parameters of pulmonary function suggests that chronic immune dysregulation may directly impact pulmonary recovery, corroborating the need for integrated immunological and pulmonary monitoring and possible therapeutic targeting in post-COVID-19 care [15,16,17]. The higher counts of CD4+/28+/192+ cells were associated with impaired TLC, suggesting chronic inflammation or delayed tissue repair may directly impact lung recovery, with the possibility of lifetime residual sequela [5,18,19].

## 6. Clinical and Public Health Implications

The high prevalence of both pulmonary and immunological sequelae in survivors of severe COVID-19 pneumonia underscores the need for comprehensive, multidisciplinary follow-up care. Although the severity of radiological abnormalities decreases over time, many patients experience persistent lung fibrosis even after 12 months, highlighting the long-term impact of severe COVID-19 on lung structure. Immune recovery is slow in patients recovering from severe and critical COVID-19, with significant declines in key immune markers persisting even at 12 months. This underscores the need for continuous monitoring of immune function in these patients. Integrating regular lung function assessments, radiological evaluations, and immunological profiling into post-discharge care plans is essential for optimising recovery. Tailored rehabilitation strategies, including physical therapy, pulmonary rehabilitation, and potentially immunomodulatory treatments, should be considered to enhance recovery and improve quality of life [19,21]. Additionally, identifying predictors of poor long-term outcomes, such as CT severity scores and specific immunological markers, can inform personalized management approaches and early interventions for high-risk patients.

## 7. Limitations

This study has several limitations that should be acknowledged. The relatively small sample size and single-centre design may limit the generalisability of our findings. Additionally, the observational nature of this study precludes causal inferences. Additionally, the lack of a control group of patients who experienced mild or moderate COVID-19 makes it difficult to determine if the observed long-term effects are specific to severe and critical cases. Lastly, while we assessed several immunological markers, other relevant factors, such as genetic predispositions, comorbidities, and environmental influences, were not controlled for, which may confound the interpretation of our results. Future research should address these limitations by incorporating larger, diverse cohorts and longitudinal study designs to better elucidate the long-term immunological effects of severe COVID-19.

## 8. Future Directions

Longitudinal studies extending beyond 12 months are needed to fully understand the trajectory and duration of pulmonary and immunological sequelae in COVID-19 survivors. Future studies should explore interventions such as pulmonary rehabilitation and immune-modulating treatments to mitigate chronic complications. Investigating the mechanisms of immune dysregulation, particularly changes in the T-cell subset, could uncover therapeutic targets for preventing pulmonary fibrosis. Identifying biomarkers and genetic risk factors may help predict which patients are at higher risk for long-term complications. Finally, comparative studies with other respiratory conditions and analysis of factors like vaccination status, age, and gender could provide further insights into personalized post-COVID-19 care.

## 9. Conclusions

Our study demonstrates that the majority of survivors of severe and critical COVID-19 experience persistent pulmonary impairments and immune dysregulation even after 12 months. While lung function improves over time, many patients continue to show residual fibrotic changes and reduced diffusion capacity. Additionally, significant reductions in CD3+, CD8+, and CD3HLA-DR+ cells were observed, indicating ongoing immune dysregulation.

This study supports a model where CD4+ T cells exacerbate pulmonary sequelae, possibly through sustained activation that impairs healing, while CD8+ T cells facilitate a resolution of inflammatory damage, leading to better functional outcomes. Elevated CD4+/28+/192+ T cells, a marker of immune dysregulation, align with more severe restrictive spirometry patterns, while higher CD8+ T cells correlate with improved lung function.

This contrast underscores the importance of CD8+ T cells in recovery and highlights CD4+ T cells’ potential role in chronic lung injury following severe COVID-19.

These findings highlight the importance of integrated post-COVID-19 care, combining pulmonary function monitoring and immunological assessments to identify patients at risk of long-term complications. Further research is needed to explore therapeutic strategies aimed at restoring immune balance and mitigating chronic lung damage, improving recovery and quality of life for these patients.

## Figures and Tables

**Figure 1 medicina-60-01954-f001:**
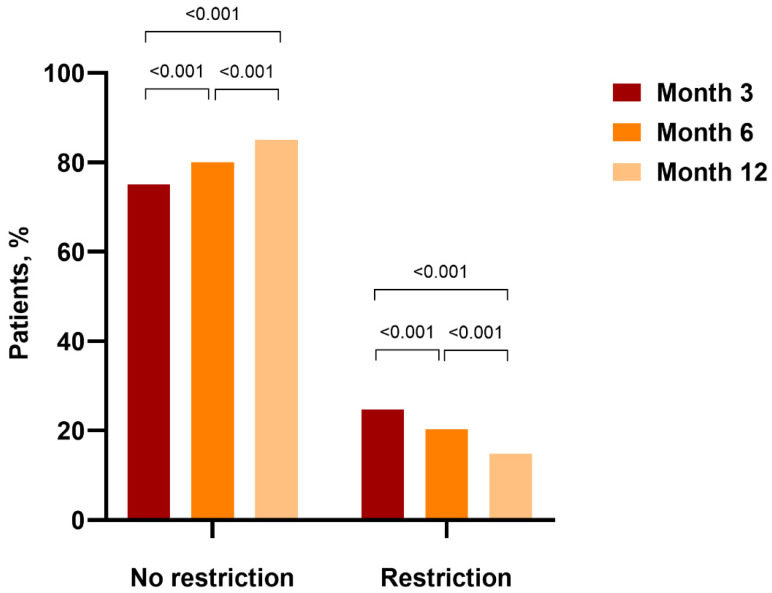
Rates of patients with and without restriction at 3, 6, and 12 months post-discharge.

**Figure 2 medicina-60-01954-f002:**
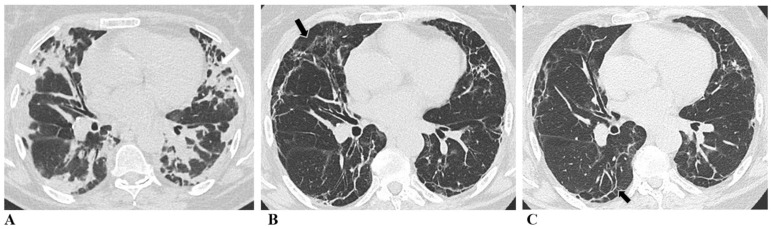
Evolution of chest CT features post-COVID-19 pneumonia. (**A**) Chest CT at initial presentation: bilateral multifocal organizing consolidation zones predominantly in the subpleural parts of the lung parenchyma (white arrows). (**B**) Six-month follow-up chest CT: in place of previously seen consolidation zones, we can see septal thickening and parenchymal bands with subtle ground glass opacities (black arrows). (**C**) Twelve-month follow-up chest CT: we can see almost complete resorption of ground glass opacities, with persistent subpleural curvilinear parenchymal bands (black arrows).

**Figure 3 medicina-60-01954-f003:**
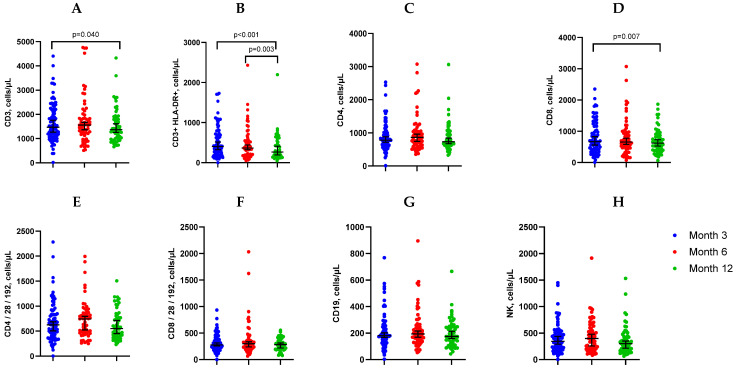
Concentrations of lymphocyte subpopulations CD3+ (**A**), CD3+HLA-DR+ (**B**), CD4+ (**C**), CD8+ (**D**), CD4/28/192+ (**E**), CD8/28/192+ (**F**), CD19+ (**G**) and NK (**H**) in patients at 3, 6, and 12 months post-discharge.

**Table 1 medicina-60-01954-t001:** Pulmonary function test results at 3, 6, and 12 months post-discharge, presented as median % of the predicted value (IQR).

Parameter	Month 3	Month 6	Month 12	p1	p2	p3
FVC	95 (85–106)	99.5 (89–111.75)	101 (88.5–112.5)	<0.001	<0.001	0.123
FEV1	96 (85.5–106)	99.5 (90.3–108.75)	103.00 (88.5–110.5)	0.005	0.002	0.032
FEV1/FVC	83 (80.5–86)	83 (78–85)	82 (79–85.5)	0.007	0.181	0.214
TLC	88 (76.5–100)	90 (81–104)	90 (80.5–102)	0.086	0.333	0.223
VC	96 (84–110)	102 (92–112.75)	103 (90.5–113)	<0.001	0.001	0.431
RV	76 (56.5–104.50)	79.5 (55.25–108.25)	75 (56.5–95.5)	0.664	0.593	0.127
DLCO	76 (64–84.5)	77 (68–86.75)	79 (71.5–84.5)	0.127	0.054	0.658

Wilcoxon Signed-Rank Test for paired samples was used for calculations; FVC—forced vital capacity, FEV1 forced expiratory capacity at the first second of exhalation, TLC—total lung capacity, VC—vital capacity, RV—residual volume, DLCO—diffusion capacity of the lung for carbon monoxide; p1—comparison of month 3 vs. month 6 test results; p2—comparison of month 3 vs. month 12 test results; p3—comparison of month 6 vs. month 12 test results.

**Table 2 medicina-60-01954-t002:** Computed tomography severity scores in patients at 6 and 12 months post-COVID-19 by lung lobe, mean ± SD.

Lung Lobe	Inflammatory Changes			Fibrotic/Reticular Changes		
	Month 6	Month 12	*p*	Month 6	Month 12	*p*
Right upper lobe	1.59 ± 1.60	1.32 ± 1.54	0.004	1.26 ± 1.22	1.01 ± 1.21	0.004
Right middle lobe	1.38 ± 1.58	1.14 ± 1.47	0.005	1.00 ± 1.18	0.90 ± 1.14	0.157
Right lower lobe	2.17 ± 1.68	1.78 ± 1.64	<0.001	1.76 ± 1.43	1.58 ± 1.45	0.012
Left upper lobe	1.42 ± 1.46	1.30 ± 1.48	0.132	1.14 ± 1.02	0.94 ± 1.01	0.005
Left lower lobe	1.96 ± 1.79	1.56 ± 1.62	<0.001	1.63 ± 1.53	1.45 ± 1.50	0.016

Wilcoxon Signed-Rank Test for paired samples was used for calculations.

**Table 3 medicina-60-01954-t003:** White blood cell, lymphocyte count, and lymphocyte subpopulation count in patients at 3, 6, and 12 months post-discharge, expressed as absolute number 10^9^/L, percentage (IQR).

Parameter	Month 3	Month 6	Month 12	p1	p2	p3
WBC	6100	6000	6000	0.871	0.613	0.527
Lymphocytes	2077 36.5 (29.25–43.75)	2107 34 (27–41.5)	1845 33 (26–40.25)	0.264	0.034	0.598
CD19+	180.15 8.5 (6–11)	193.92 9 (7–12)	174.42 9 (6–12.3)	0.392	0.291	0.267
CD4+	778.95 38 (31.25–44)	862.84 40 (33–44.5)	729.28 40 (35–46)	0.909	0.284	0.413
CD8+	665.20 31 (23.25–40.75)	653.17 30 (21–38.5)	618.64 29 (22–39)	0.487	0.007	0.174
CD3+	1465.93 70 (65–76.5)	1555.2 72 (64.5–75)	1361.6 72 (66–78)	0.774	0.040	0.163
NK	344.76 16 (11–24.5)	397.60 16 (11–23.75)	300.72 14 (10–22)	0.466	0.108	0.460
CD3HLA-DR+	403.62 20.5 (14–26.75)	367.54 17 (12–22.5)	265.27 15 (10–22)	0.070	<0.001	0.003
CD4+/28+/192+	625.24 27.7 (22–35.1)	745.18 28.9 (23.5–38)	550.37 27.98 (22.5–35.12)	0.927	0.103	0.195
CD8+/28+/192+	276.64 13.2 (9.55–17.75)	297.62 13.55 (10.38–16.97)	280.29 13.70 (9.90–17.3)	0.775	0.353	0.433

Wilcoxon Signed-Rank Test for paired samples was used for calculations; p1—comparison of month 3 vs. month 6 test results; p2—comparison of month 3 vs. month 12 test results; p3—comparison of month 6 vs. month 12 test results.

**Table 4 medicina-60-01954-t004:** Computed tomography severity scores in patients at 3, 6, and 12 months post-COVID-19, grouped by restrictive spirometry pattern, median (IQR).

CT Severity Score	Restriction	No Restriction	*p*
Month 3			
Total score during acute COVID-19	17 (12–21)	15 (8.5–19)	0.140
Month 6			
Inflammatory changes at 6 months	10.5 (3.25–18.75)	7 (0.25–13)	0.129
Fibrotic/reticular changes at 6 months	9.5 (4–12.5)	5 (1.25–11)	0.090
Month 12			
Inflammatory changes at 12 months	10 (1–15)	4 (0–11)	0.211
Fibrotic/reticular changes at 12 months	11 (3–13)	4 (0–10)	0.203

Mann–Witney U Test was used for calculations; CT—computed tomography.

**Table 5 medicina-60-01954-t005:** Association of pulmonary function tests, radiological findings, and immunological markers in patients at 3, 6, and 12 months post-COVID-19.

Parameter	Predictors	β (95% CI)	*p*
Month 3			
FEV1/FVC	CD8/28/192	−0.01 (−0.02–−0.001)	0.035
DLCO	Lymphocytes	0.06 (0.02–0.11)	0.008
	NK	−0.06 (−0.10–−0.01)	0.021
	Female gender	−8.46 (−15.33–−1.59)	0.016
Month 6			
TLC	CD4/28/192	−0.02 (−0.05–−0.004)	0.020
DLCO	Age	−0.67 (−1.14–−0.19)	0.007
	Female gender	−10.23 (−18.86–−1.61)	0.021
Month 12			
FVC	CD4/28/192	−0.07 (−0.10–−0.04)	0.001
	CD8/28/192	0.10 (0.04–0.16)	0.002
FEV1	CD3HLA-DR+	−0.04 (−0.08–−0.004)	0.031
	CD4/28/192	−0.06 (−0.09–−0.03)	0.001
	CD8/28/192	0.08 (0.01–0.15)	0.023
TLC	CD8/28/192	0.08 (0.02–0.14)	0.015
VC	CD3HLA-DR+	−0.04 (−0.07–−0.004)	0.028
	CD4/28/192	−0.06 (−0.09–−0.04)	<0.001
	CD8/28/192	0.1 (0.04–0.16)	0.002
RV	CD8	0.20 (0.03–0.36)	0.024
DLCO	Female gender	−10.16 (−18.45–−1.87)	0.017
Inflammatory changes	Age	0.22 (0.01–0.43)	0.038
Fibrotic/reticular changes	Age	0.18 (0.01–0.35)	0.044

Multivariable linear regression was used for calculations, and only statistically significant associations are included in the table. FVC—forced vital capacity, FEV1—forced expiratory capacity at the first second of exhalation, TLC—total lung capacity, VC—vital capacity, RV—residual volume, DLCO—diffusion capacity of the lung for carbon monoxide.

## Data Availability

The original contributions presented in the study are included in the article, further inquiries can be directed to the corresponding authors.

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
