# Peer review of "Long-Term Pulmonary Sequelae and Immunological Markers in Patients Recovering from Severe and Critical COVID-19 Pneumonia: A Comprehensive Follow-Up Study"

_medicina, 2024, doi:10.3390/medicina60121954_

Round 1

Reviewer 1 Report

Comments and Suggestions for Authors

Background

1.Line 75: “However, in severe COVID-19, these cells often become dysfunctional, contributing to the prolonged inflammation and immune dysregulation seen in long COVID.” reference should be inserted for this conclusion. It cannot be a logical deduction of those stated before.

Material and methods

2.Line 114: “from October 2021 to October 2022” Why this period was used? It is linked somehow to a virus variant in your Hospital? (Omicron B types? BQ and BA), maybe an explanation would be appreciated.

3.It is not very clearly described the pool of the patients was taken into the study. There were evaluated all patients in the mentioned period with Covid positive (PCR) and selected those with severe and critical infection? Those 85 patients are the results of some work, or it was a database in the Hospital HIS where the severity of Covid was selected?

4.Line 186: reference error

5.Line 204: “The results were evaluated by experienced immunologist”. And? How was quantified the result?

6.What were the exclusion criteria? Patients should survive until 12 months after discharge? No patients has died in hospital? How many were excluded? Or none? The study it is prospective, interventional one or just retrospective observational? Patients with pulmonary infections or Covid re-infection after discharge were excluded?

Results

7.Line 255 to 276: “A total of 59 out of 76 …, A total of 55 out of 79” What are those total number of patients? 76, 79, 76, 78? Should be described at results, why not a total of 85?

8. Table 2: values in brackets are IQR or minim and maxim? Should be mentioned somewhere to be clear for everyone.

9.Table 2: for same values, IQR as well, p values are statistically significant (for example fibrosis first line), I understand the statistical calculations behind that, but it seems weird a little. Those are median values? Should be specified. Maybe using the average values instead of median, it may improve it? What is your opinion.

10.Table 3. I recommend that the N number should be addressed, why there are not 85 patients for all series. Personally, I consider that the Table 1 and 2 are more readable than Table 3. There are so many parameters which are redundant somehow, absolut number, percentage, IQR twice for them, minimal and maximal value. I would use absolute number, percentage and IQR of percentage. In this way maybe the table can be arranged similar to Table 1, p number also should be included.

11.Lines 312-320, 351-355 and 380-385: the statistical important results are described, I wouldn’t recommend repeating all values, just one number and p value if you do not decide to include p values in the table (IQR is not relevant here).

Conclusions

12.Line 510-511: “Our study demonstrates that survivors of severe and critical COVID-19 experience 510 persistent pulmonary impairments and immune dysregulation even after 12 months” After this statement it looks like all of them have pulmonary impairments. We know that were patients without any pulmonary affection. Majority or many of them term should be used here to describe how many of the patients got pulmonary sequelae.

Author Response

Thank You for Your detailed analysis and helpful questions and suggestions. We hope that the answers and corrections made the article more precise.

Comments and Responses: 

1.Line 75: “However, in severe COVID-19, these cells often become dysfunctional, contributing to the prolonged inflammation and immune dysregulation seen in long COVID.” reference should be inserted for this conclusion. It cannot be a logical deduction of those stated before.

Response: Thank You for the absolutely right comment,- reference is added.

2.Line 114: “from October 2021 to October 2022” Why this period was used? It is linked somehow to a virus variant in your Hospital? (Omicron B types? BQ and BA), maybe an explanation would be appreciated.

Response: The follow up study was not correlated with circulating virus type, simply the beginning of the study was innitiated just after we received the permission of the Ethical Committee and hospital administration; and organised the visitsts to pulmonologist with CT scans, pulmonary functions testing; and received the laboratorial material for blood immunologocal parameters analysis.

3.It is not very clearly described the pool of the patients was taken into the study. There were evaluated all patients in the mentioned period with Covid positive (PCR) and selected those with severe and critical infection? Those 85 patients are the results of some work, or it was a database in the Hospital HIS where the severity of Covid was selected?

Response: Thank You for the question. All patients in the mentioned period with Covid positive (PCR) with severe and critical disease treated in our hospital at the time of discharge were offered to participate in this study (if they met the inclusion criteria). This was a prospective study. I made the corrections in the artitle, hoping to make it more precise.

4.Line 186: reference error

Response: I made the corrections using EndNote.

5.Line 204: “The results were evaluated by experienced immunologist”. And? How was quantified the result?

Response: Thanks for Your comment. I deleted this part, possibly it was not necessary to emphasize, as the analysis and interpretation of the immunologist are presented in results and discussion.

6.What were the exclusion criteria? Patients should survive until 12 months after discharge? No patients has died in hospital? How many were excluded? Or none? The study it is prospective, interventional one or just retrospective observational? Patients with pulmonary infections or Covid re-infection after discharge were excluded?

Response: Thank You for the helpful question. I corrected the Methodology , as mentioned in the question 3. This was a prospective study and only survivals were included at the time of discharge from the hospital. Thewe were no any patients deaths or severe COVID re-infections ( or other infections) during the observational period.

7.Line 255 to 276: “A total of 59 out of 76 …, A total of 55 out of 79” What are those total number of patients? 76, 79, 76, 78? Should be described at results, why not a total of 85? 

Response: Thank You for Your comment. Not all patients agreed to undergo repeat CT scans at 6 and 12 months, and we also experienced some supply issues that affected the completion of immunological markers testing for all patients. To clarify, we have included a description of the total number of patients evaluated at the beginning of each results section for better transparency.  

  1. Table 2: values in brackets are IQR or minim and maxim? Should be mentioned somewhere to be clear for everyone. 

Response: Thank You for Your helpful suggestion. We have revised the title of Table 2 to specify that values are reported as severity scores mean and standard deviation. The updated title now reads: 'Computed tomography severity scores in patients at 6, and 12 months post-COVID-19 by lung lobe, mean ± SD.”  

9.Table 2: for same values, IQR as well, p values are statistically significant (for example fibrosis first line), I understand the statistical calculations behind that, but it seems weird a little. Those are median values? Should be specified. Maybe using the average values instead of median, it may improve it? What is your opinion. 

Response: Thank You for Your thoughtful comment. To address the potential confusion regarding the presentation of statistically significant p-values for the same median values, we have revised the Table 2 to report the mean with standard deviation instead of the median and IQR. We believe this change enhances the clarity and interpretability of the results. 

10.Table 3. I recommend that the N number should be addressed, why there are not 85 patients for all series. Personally, I consider that the Table 1 and 2 are more readable than Table 3. There are so many parameters which are redundant somehow, absolute number, percentage, IQR twice for them, minimal and maximal value. I would use absolute number, percentage and IQR of percentage. In this way maybe the table can be arranged similar to Table 1, p number also should be included. 

Response: Thank You for Your valuable feedback. We have revised Table 3 to improve readability by keeping only the absolute numbers, percentage, and IQR for percentage, as you suggested. We have also reorganized the table to present results in separate columns for months 3, 6, and 12, and added p-values. The revision aligns Table 3 more closely with the format of Tables 1 and 2. We have also added a description of the totalnumber of patients for whom immunological markers were tested at the beginning of the results section. 

11.Lines 312-320, 351-355 and 380-385: the statistical important results are described, I wouldn’t recommend repeating all values, just one number and p value if you do not decide to include p values in the table (IQR is not relevant here). 

Response: Thank You for Your suggestion. We have revised the text in Lines 312-320, 351-355, and 380-385 to avoid repeating values and have removed the IQR. We now report only the key result and the p-value, as recommended.2.Line 510-511: “Our study demonstrates that survivors of severe and critical COVID-19 experience 510 persistent pulmonary impairments and immune dysregulation even after 12 months” After this statement it looks like all of them have pulmonary impairments. We know that were patients without any pulmonary affection. Majority or many of them term should be used here to describe how many of the patients got pulmonary sequelae

12.Line 510-511: “Our study demonstrates that survivors of severe and critical COVID-19 experience 510 persistent pulmonary impairments and immune dysregulation even after 12 months” After this statement it looks like all of them have pulmonary impairments. We know that were patients without any pulmonary affection. Majority or many of them term should be used here to describe how many of the patients got pulmonary sequelae.

Response: Thank You for very important and useful suggestion. I corrected this part to :"Our study demonstrates that majority of survivors of severe and critical COVID-19 experience persistent pulmonary impairments and immune dysregulation even after 12 months".

Reviewer 2 Report

Comments and Suggestions for Authors

The work presented for review concerns an important topic. This prospective observational cohort study aimed to assess lung function, radiological changes, and some immune parameters in survivors of severe and critical COVID-19 up to 12 months after hospital discharge. The research is interesting and presented correctly. I have some minor comments as below.

1.       If available, provide baseline characteristics of the patients with COVID-19, laboratory and clinical findings at hospital admission, please.

2.       Line 27, proffibrrotic. Correct, please.

3.       Lines 73-74, “T lymphocytes, especially T cells,…”. Do you mean “The lymphocytes, especially T cells,…”?

4.       Lines 186-187, [Error! Reference source not found.,Error! Reference source not found.]. Correct, please.

5.       The results are duplicated:

At 3 months, 24.71% of patients showed restrictive pattern in spirometry, with the rate statistically significantly decreasing to 20.24% at 6 months and 14.8% at 12 months (Figure 1) (lines 247-248).

Restrictive spirometry pattern was observed in 21 out of 85 (24.71%) patients at 3 months, in 17 out of 84 (20.24%) patients at 6 months and in 12 out of 81 (14.81%) of patients at 12 months (lines 336-337).

6.       Line 194, What do you mean by CD16+56?

7.       Please, make CD writing uniform, including figures and tables. Sometimes “+” is missing.

8.       Were there age and sex differences between the patients with and without restriction at 3, 6, and 12 months post-discharge?

Author Response

Dear Reviewer, thank You for the detailed analysis. We made very useful corrections based on Your questions and notifications. I believe these corrections made the article more precise.

  1. If available, provide baseline characteristics of the patients with COVID-19, laboratory and clinical findings at hospital admission, please.

Response:  Thank You for the question. We decided not to analyse laboratory/clinical findings at the admission in this study as far as we performed a follow-up and in this study we do not have the data for the observational correlations. Also, the article includes a lot of parameters in the analysis therefore we decided that more data would overload the article and it would become uncomfortable to read and analyse. I agree that these data could be analysed and might useful, may be it could become another topic for a next article.I sincerely hope my response is acceptable.

  1. Line 27, proffibrrotic. Correct, please.

Response: Thank You for noticing. Corrected in the text.

  1. Lines 73-74, “T lymphocytes, especially T cells,…”. Do you mean “The lymphocytes, especially T cells,…”?

Response: Thank You for noticing. Corrected in the article.

  1. Lines 186-187, [Error! Reference source not found.,Error! Reference source not found.]. Correct, please.

Response: Thank You for noticing. Corrected using EndNote.

  1. The results are duplicated:

At 3 months, 24.71% of patients showed restrictive pattern in spirometry, with the rate statistically significantly decreasing to 20.24% at 6 months and 14.8% at 12 months (Figure 1) (lines 247-248).

Restrictive spirometry pattern was observed in 21 out of 85 (24.71%) patients at 3 months, in 17 out of 84 (20.24%) patients at 6 months and in 12 out of 81 (14.81%) of patients at 12 months (lines 336-337).

Response: Thank You for noticing. Repeated part deleted in the article.

  1. Line 194, What do you mean by CD16+56?

Response. Thank You for noticing. Deleted in the text.

  1. Please, make CD writing uniform, including figures and tables. Sometimes “+” is missing.

Response: Thank You for the suggestion. We were debating whether we should use “+” in the tables or not (in the literature we found variations), but I think You are right, it should be the same in all the text, tables and figures. CD was uniform was corrected to be identical everywhere

  1. Were there age and sex differences between the patients with and without restriction at 3, 6, and 12 months post-discharge?

Response: Thank you for the question. There were no statistically significant differences in age or sex between the patients with and without restriction at 3, 6, and 12 months post-discharge.

Reviewer 3 Report

Comments and Suggestions for Authors

i. The roles of CD192 were described in the introduction, however, the molecule was not part of the study parameters (study objectives and rationale). Clarify.

ii. Exclusin criteria: Since the study also investigated the patients’ immune functions post-discharge, do the authors think if the subjects already suffered from immune disorders before COVID-19 but were still included in this study would affect the result interpretations?

iii. Line 158: Do you mean the subjects received posterior-anterior and lateral chest X-rays in the first visit?  

iv. Lines 186-187: Reference not found?

v. Lines 255-275: It is suggested to tabulate the results to help readers comprehend the findings.

vi. Line 276: Replace the term “example” with “representatives”.

vii. Table 3: What do the N numbers decrease over time?

viii. Lines 427-434: Deliberate how the greater level of CD4+ T cells is associated with the exacerbation of pulmonary issues in comparison to CD8+ T cells. 

Author Response

Dear Reviewer,

Thank You for the detailed analysis of our article. The comments and questions helped us to clarify and more complement the submitted information.

These are the answers and corrections that were made:

1. The roles of CD192 were described in the introduction, however, the molecule was not part of the study parameters (study objectives and rationale). Clarify.

Response: Thank You for the question. In this study we included the analysis of this molecule having CD4 and CD8 cells: these data are presented in CD8+/28+/192+ and CD4+/28+/192+ cell counts and associations with pulmonary function in the parts of Results and Discussion.

2. Exclusin criteria: Since the study also investigated the patients’ immune functions post-discharge, do the authors think if the subjects already suffered from immune disorders before COVID-19 but were still included in this study would affect the result interpretations?

Resposnse: Thank You for this useful question and comment. There were no immunocompromised patients included in this study. We corrected the inclusion criteria in the article to avoid misunderstanding.

 3. Line 158: Do you mean the subjects received posterior-anterior and lateral chest X-rays in the first visit?  

Response: Thank You for the question. Some patients had only X-Rays at the time of admission, but the majority underwent CT-scans to dismiss the pulmonary artery thromembolic complications. During follow-up visit X-Rays were not performed, only CT analyses were used.

 4. Lines 186-187: Reference not found?

Response: Thank You for noticing – corrected using EndNote.

 5. Lines 255-275: It is suggested to tabulate the results to help readers comprehend the findings.

Response: Thank You for the suggestion. There are radiological data presented in Table 2 (Computed tomography severity scores in patients at 6, and 12 months post-COVID-19 by lung lobe). Also, the CT scores and pulmonary function correlation is also presented in Table4. There are quite many tables in the article, we think it would overload the text with more tables, as the main findings are already presented in these 2 tables.

 6. Line 276: Replace the term “example” with “representatives”.

Response: Thank You for a useful suggestion. Changed in the article.

 7. Table 3: What do the N numbers decrease over time?

Response: Thank you for your question. Not all patients agreed to undergo repeat CT scans at 6 and 12 months, and we also experienced some supply issues that affected the completion of immunological markers testing for all patients. To clarify, we have included a description of the total number of patients evaluated at the beginning of each results section for better transparency.

8. Lines 427-434: Deliberate how the greater level of CD4+ T cells is associated with the exacerbation of pulmonary issues in comparison to CD8+ T cells. 

Response:  Thank You for a very useful question and comment. We corrected the article with this additional information

“The study indicates that elevated levels of CD4+ T cells, particularly the CD4+/28+/192+ subset, correlate with poorer lung function outcomes post-COVID-19. These cells are markers of immune activation and exhaustion. Increased CD4+ T cell counts are associated with reduced  TLC and decreased FVC and FEV1. The persistence of high CD4+ T cell counts suggests ongoing inflammation or delayed lung tissue repair, which could contribute to structural lung changes and the exacerbation of respiratory impairment. 

Conversely, CD8+ T cells appear beneficial for lung function recovery. Elevated CD8+/28+/192+ counts were positively associated with improved pulmonary parameters, such as FVC, FEV1, and VC. This suggests a potentially protective role of CD8+ T cells in mitigating fibrosis or inflammatory damage. CD8+ T cells likely contribute to a balanced immune response, aiding in the resolution of inflammation without inducing excessive fibrosis or structural changes in lung tissues.

And also added the informations at the Conclusions:

“The study supports a model where CD4+ T cells exacerbate pulmonary sequelae, possibly through sustained activation that impairs healing, while CD8+ T cells facilitate a resolution of inflammatory damage, leading to better functional outcomes. Elevated CD4+/28+/192+ T cells, a marker of immune dysregulation, align with more severe restrictive spirometry patterns, while higher CD8+ T cells correlate with improved lung function.

This contrast underscores the importance of CD8+ T cells in recovery and highlights CD4+ T cells’ potential role in chronic lung injury following severe COVID-19”.

Round 2

Reviewer 1 Report

Comments and Suggestions for Authors

Can be accepted in present form.